# Transcriptome Analysis of Caco-2 Cells upon the Exposure of Mycotoxin Deoxynivalenol and Its Acetylated Derivatives

**DOI:** 10.3390/toxins13020167

**Published:** 2021-02-22

**Authors:** Yuyun He, Xiaoyao Yin, Jingjing Dong, Qing Yang, Yongning Wu, Zhiyong Gong

**Affiliations:** 1Key Laboratory for Deep Processing of Major Grain and Oil of Ministry of Education, Wuhan Polytechnic University, Wuhan 430023, China; Heyuyun1997@163.com (Y.H.); yxy186139@163.com (X.Y.); DongJingjingWHQG@163.com (J.D.); qingyangwhpu@yeah.net (Q.Y.); 2China National Center for Food Safety Risk Assessment, NHC Key Laboratory of Food Safety Risk Assessment, Food Safety Research Unit (2019RU014) of Chinese Academy of Medical Science, Beijing 100000, China; wuyongning@cfsa.net.cn

**Keywords:** deoxynivalenol, 3-acetyldeoxynivalenol, 15-acetyldeoxynivalenol, Caco-2, RNA-seq

## Abstract

Deoxynivalenol (DON), 3-acetyldeoxynivalenol (3-ADON) and 15-acetyldeoxynivalenol (15-ADON) are type B trichothecenes; one of the major pollutants in food and feed products. Although the toxicity of DON has been well documented, information on the toxicity of its acetylated derivative remains incomplete. To acquire more detailed insight into 3-ADON and 15-ADON, Caco-2 cells under 0.5 µM DON, 3-ADON and 15-ADON treatment for 24 h were subjected to RNA-seq analysis. In the present study, 2656, 3132 and 2425 differentially expressed genes (DEGs) were selected, respectively, and were enriched utilizing the Kyoto Encyclopedia of Genes and Genomes (KEGG) and the Gene Ontology (GO) database. The upregulation of ataxia-telangiectasia mutated kinase (ATM), WEE1 homolog 2 (WEE2) and downregulation of proliferating cell nuclear antigen (PCNA), minichromosome maintenance (MCMs), cyclin dependent kinase (CDKs), and E2Fs indicate that the three toxins induced DNA damage, inhibition of DNA replication and cell cycle arrest in Caco-2 cells. Additionally, the upregulation of sestrin (SENEs) and NEIL1 implied that the reason for DNA damage may be attributable to oxidative stress. Our study provides insight into the toxic mechanism of 3-ADON and 15-ADON.

## 1. Introduction

Mycotoxins are toxic secondary metabolites produced by a few fungal species belonging mainly to the *Aspergillus*, *Penicillium*, *Fusarium* and *Alternaria* genera [1]. They are commonly found in cereals, fruits, and spices [2]. Although there are hundreds of mycotoxins known today, only a few of them have aroused widespread attention [3]. Among mycotoxin groups, trichothecenes are of increasing concern with respect to food safety issues due to their frequent and global occurrence [4]. Trichothecenes can be classified into four groups (A–D) based on their chemical structures, type A and type B trichothecenes are predominant in food and feed [5]. Deoxynivalenol (DON), is one of the main representatives of type B trichothecenes [6,7,8]. Due to its thermal stability and high resistance to processing, DON can survive most food processing treatments, and thus, it can be frequently found in cereal commodities (i.e., cornflakes, biscuit and wheat bran) intended for animal or human consumption [9,10,11].

JECFA (Joint FAO/WHO Expert Committee on Food Additives) established the PMTDI (Provisional Maximum Tolerable Daily Intake) for DON as 1 μg/kg bw per day in 2011 [12]. However, in recent years, reports have evidenced more frequently detecting more than one type of mycotoxin in food and feed due to the ability of most *Fusarium* species to simultaneously produce different mycotoxins [13]. In addition, plants and some kinds of microorganisms infected with mycotoxin-producing fungi have the ability to—through their defense mechanisms—alter the structures of DON, transforming it into deoxynivalenol-3-glucoside (D3G), 3-acetyldeoxynivalenol (3-ADON), 15-acetyldeoxynivalenol (15-ADON) and DON-sulfates [2,14]. During physical processes such as baking, DON was found to degrade into isoDON, norDON B and norDON C [15]. In mammals, DON can be converted into deoxynivalenol-3-glucuronide (DON-3-GlcA) and deoxynivalenol-15-glucuronide (DON-15-GlcA) through liver microsomes [16]. For DON, 3-ADON and 15-ADON, the mean amounts of these compounds in Belgian cereal-based food products were found to be 42, 25 and 16 µg/Kg, respectively [17]. Zhao et al. analyzed 30 mycotoxins in animal feed (swine feed and poultry feed) and found that out of 30 feed samples, 28 samples (93%) were contaminated with at least two toxins. The most frequently occurring mycotoxins were fumonisins, zearalenone, DON, 3-ADON, and 15-ADON [18]. Palacios et al. investigated the occurrence of DON, 3-ADON, 15-ADON and DON-3-GlcA in 84 durum wheat samples from Argentinean, and found that all samples were positive for DON (on average detecting an amount of 1750 μg/kg), 94% were positive for D3G and 49% were positive for acetylated derivates of DON [19]. In another study assessing different cereals and cereal-derived food in Ika, Werke and Moulinex showed that the occurrence of DON was 77% and the acetylated forms, 3-ADON and 15-ADON were found to occurred in 87 and 73% of the samples, respectively [20]. Mastanjević et al. found that DON, its modified plant metabolite D3G, brevianamide F, tryptophol, linamarin, lotaustralin, culmorin (CUL), 15-hydroxy-CUL and 5-hydroyx-CUL were present in all malt and beer samples in Osijek [21]. It is noteworthy that DON was found in infants and young children consuming household- and industrially-processed complementary foods in Nigeria [22]. Even though the acetylated derivatives of DON are, in general, less frequently found and constitute less than 10% of the total DON concentrations, they are considered to be as toxic as DON. In addition, 3-ADON and 15-ADON are converted to DON in vivo and in vitro, therefore, these compounds can contribute to the total DON-induced toxicity [23]. Furthermore when a combination of 3-ADON + 15-ADON is found in HepG2 cells, a synergistic effect can be observed at 24, 48, and 72 h [11]. In the intestinal epithelial cell line porcine intestinal epithelial cells-1 (IPEC-1), the combination of DON and 3-ADON or 15-ADON showed the same synergistic effects after 24 h [17]. Because of the serious concerns for animal and human health, regulatory or guideline levels for DON and its derivatives in food and feed are in force in more than 40 countries [24]. JECFA determined the PMTDI for DON to be a total PMTDI of 1 μg/kg bw per day for DON and its acetylated derivatives (3-ADON and 15-ADON) [5].

The toxicology of DON is well documented. Acute exposure of experimental animals to extremely high doses of DON in feeds causes mortality or marked tissue injury, whereas acute exposure to relatively low doses induces vomiting in animals, especially pigs. After consumption of scabby wheat containing high levels of DON, acute human illness with symptoms similar to gastroenteritis were observed [25]. It may also induce abdominal pain, increased salivation, diarrhea and emesis in humans [24]. Chronic exposure of pigs to 2 ppm of DON restricts growth and disrupts immune functioning [26]. At the molecular level, the toxicological effects of DON are based on the interaction of its 12–13 epoxide moiety with the A-site of the eukaryotic 60S ribosomal subunit, generating so-called ribotoxic stress and interfering with the elongation step of protein formation [11,27]. Reports that utilized transcriptome analysis to explore the damage of DON have demonstrated that: deleterious pro-inflammatory and oxidative stress can be observed in pig jejunum segments; ribosomes, protein synthesis, and endoplasmic reticulum stress (ER stress) can be affected in the human T lymphocyte cell line Jurkat; and human peripheral blood mononuclear cells, cell cycle arrest and apoptosis can be induced in human chondrocytes [28,29,30]. Zhang et al. demonstrated that, in HepG2 cells, DON facilitated the production of ROS, which in turn induced DNA damage and led to cell cycle arrest and apoptosis [31].

Although far less common than studies of DON, recently, an increasing number of studies related to the damaging effects of 3-ADON and 15-ADON have been documented. The toxicological in vitro effects of 3-ADON and 15-ADON have been investigated in the context of the immune system, intestinal issues, oxidative stress and cell cycle in several studies [7,11,13,32]. The cytotoxicity of the three toxins in mammalian intestinal cells ranked in the order of 15-ADON > DON > 3-ADON, with similar results found for barrier function, MAPkinase activation and expression of tight junctions, and histological alterations. Estimates of mean acute and chronic exposure to different age groups from European countries to DON, 3-ADON, 15-ADON and D3G range from 0.2 to 2.9 and 0.2 to 2.0 μg/kg body weight (bw) per day. The European Food Safety Authority (EFSA) panel on contaminants in the food chain identified vomiting as a critical acute effect to characterize the acute hazard of the sum of DON, 3-ADON, 15-ADON and D3G in humans. Currently, available data for the single or combined genotoxicity and/or carcinogenicity of 3-ADON and 15-ADON are limited [33]. In order to provide a comprehensive and detailed insight into the toxicological alterations induced by DON and its acetylated derivates, we used Caco-2 cells—as it has been demonstrated that they possess identical cell polarity, and similar tight junction and pinocytosis functioning to human intestinal epithelial cells—to analyze the changes that occur in the transcriptome [3].

## 2. Results

### 2.1. 3-ADON Exerts Less Harmful Effects Than DON and 15-ADON on Cultured Caco-2 Cells

The cytotoxic effects induced by DON, 3-ADON and 15-ADON on cell viability were evaluated. Caco-2 cells were exposed to different concentrations (0.1, 0.2, 0.5, 1, 2, 5, 10 µM) of DON, 3-ADON and 15-ADON, and treated for 24, 48, 72 h, respectively. The cell viability of Caco-2 cells shows a time- and dose-dependent decrease. As shown in Figure 1, DON, 3-ADON and 15-ADON reduced cell viability at higher tested concentrations. The cell viability measurements after exposure to DON, 3-ADON and 15-ADON at 24 h were 55.94, 60.49 and 48.86%, respectively. From 0.2 to 10 µM, the viability of Caco-2 cells under DON and 15-ADON treatment was significantly reduced. The viability of the 3-ADON treated group sharply reduced from 1 to 10 µM. Under the 48 h treatment, cell viabilities decreased to 28.50, 36.28 and 33.49%. The downtrends of DON, 3-ADON and 15-ADON were similar to the 24 h treatment. After 72 h of treatment, the viabilities of Caco-2 cells were 13.61, 22.33 and 13.07%. The robust decrease in cell viabilities occurred from 0.2 to 10 µM under DON treatment, 0.5 to 10 µM under 15-ADON treatment and 2 to 10 µM under 3-ADON treatment. The IC_50_ of values (Table 1) obtained at 24, 48 and 72 h ranged from 1.46 ± 0.42 to 6.17 ± 0.93 µM for DON, from 7.85 ± 2.01 to 13.19 ± 0.71 µM for 3-ADON, and from 1.44 ± 0.67 to 3.86 ± 0.81 µM for 15-ADON. According to the results, we can draw the conclusion that the order of potential cytotoxicity in Caco-2 cells from 0.2 to 10 µM was: 3-ADON < DON ≈ 15-ADON. The same order was observed under 0.1 µM treatment at 24 and 72 h, however, at 48 h, the order of cytotoxicity was: DON < 3-ADON < 15-ADON.

### 2.2. Transcriptomic Changes of Caco-2 Cells

In order to find out the toxic effects of DON, 3-ADON and 15-ADON, RNA-seq was used for transcriptomic analyses. After exposure of Caco-2 cells to 0.5 µM DON, 3-ADON and 15-ADON for 24 h, RNA was isolated and subjected to RNA-seq analysis. The concentration of 0.5 µM DON value corresponds to the mean estimated daily intake of French adult consumers on a chronic basis. It is considered a realistic value that the human gut is exposed to [1]. The incubation time of the cells with toxins for 24 h reflects a realistic estimate of the human gut being exposed to mycotoxins by food consumption [3]. When compared to control samples, at least 2425 genes were selected as differentially expressed genes (DEGs) in each group by applying the criteria ≥ two-fold change and adjusted *p*-value ≤ 0.01. The proportion showed no significant differences between the upregulated genes and downregulated genes. As shown in the heatmap (Figure 2), the three mycotoxins exerted a significant influence on the transcriptome of Caco-2 cells. Clustering indicated that control group separated from toxins treated groups. A Venn diagram (Figure 3) was constructed to show the relationship between DEGs of different groups. Among 2656, 3132 and 2425 DEGs under DON-, 3-ADON- and 15-ADON-treated conditions, respectively, 1688 identical genes were observed in all three groups, accounting for over 50% of each group.

### 2.3. Function Enrichment Analysis of Differentially Expressed Genes (DEGs)

To further investigate the deleterious effects of DON, 3-ADON and 15-ADON, the DEGs were subjected to pathway enrichment analysis utilizing Kyoto Encyclopedia of Genes and Genomes (KEGG) and Gene Ontology (GO) database. All the pathways significantly affected in Caco-2 cells were selected using a cut-off threshold of *p*-value ≤ 0.05 and *q*-value ≤ 0.2 in the KEGG enrichment. The DEGs of Caco-2 cells under DON, 3-ADON and 15-ADON treatment were enriched in 27, 42 and 28 pathways, respectively (Figure 4). Among these, a total of 20 signaling pathways were found to be regulated by all three mycotoxins, which suggests that their toxic effects on Caco-2 cells are partially shared. Of the pathways regulated by DON, 3-ADON and 15-ADON, three of the top five pathways are the same, including DNA replication, base excision repair and cell cycle (Table 2).

In the results of the KEGG enrichment analysis, cell cycle arrest was observed to be induced by all three toxins. At least 30 genes were identified in the analysis, and all phases of the cell cycle were involved. For 15-ADON, four genes were upregulated (WEE2, STAG1, ATM, CDKN2B), other genes (i.e., PCNA, MCM3, E2F1, CDK1, CDC45) were downregulated. Parts of these genes (i.e., ATM, CHEK1, CHEK2) which respond to DNA damage, were also enriched in the p53 signaling pathway. This suggests that the cell cycle arrest, especially in the G1 and S-phase, elicited by DON, 3-ADON and 15-ADON is partially ascribed to the changes in the p53 signaling pathway.

The DNA replication pathway was identified as the most enriched pathway for all the three toxins. Of the 20 DNA replication genes identified in the 15-ADON group (i.e., PCNA, MCM2, POLD1, POLE2, RFC4, RPA3), all of them were downregulated, and furthermore, all of these genes were also downregulated by DON and 3-ADON. The suppression of DNA replication is also in congruence with the repression of the S-phase in the cell cycle. Genes correlated with DNA biosynthesis (PCNA, MCMs) overlap with parts of the genes identified in the cell cycle. In line with these findings, base excision repair, mismatch repair and nucleotide excision repair signaling pathways, which are closely related to DNA replication, were found to be downregulated in all three groups. These four pathways were jointed affected by several genes encompassing PCNA, POLD1, POLD2, POLD3 and LIG1. The interaction between these pathways indicated a network response against DON, 3-ADON and 15-ADON.

Other pathways enriched by KEGG, including drug metabolism, pyrimidine metabolism, ascorbate and aldarate metabolism and the retinol metabolism pathway, are interrelated with each other in terms of their functions in endogenous metabolism.

The results of the GO pathways were selected using the criteria *p*-adjusted ≤ 0.01 and *q*-value ≤ 0.05. According to the GO terms, most of the pathways were enriched in biological process. Similar to the results of KEGG, the pathways which were affected significantly in Caco-2 cells have a close relationship with DNA replication and biosynthesis & metabolic process. In each of the conditions, the top 2 signaling pathways are “DNA replication” and “DNA-dependent DNA replication” (Figure 5). Both KEGG and GO analysis suggest that DON, 3-ADON and 15-ADON induced DNA damage on Caco-2 cells.

To verify the RNA-seq data, thirteen genes, comprising MCM3, MCM4, PCNA, POLE2, POLD2, RPA3, RAD51, BID, ATM, CDKN2C, E2F2, CCND2, SERPINE1, were selected for RT-PCR analysis. As shown in Figure 6, the expression of the thirteen selected genes is concordant with the results of the RNA-seq analysis, confirming the validity of the RNA-seq results.

## 3. Discussion

The frequent and global occurrence of DON and its acetylated derivatives in cereal-based products has aroused worldwide attention. Although the toxic effects of DON have been well characterized, the toxicity of 3-ADON and 15-ADON remains unclear. Thus, the present study sheds further light on the deleterious effects that DON, 3-ADON and 15-ADON exert on the transcriptomes of Caco-2 cells in vitro.

Firstly, cell viability experiments were performed and the results show that the viability of Caco-2 cells under DON, 3-ADON and 15-ADON treatments was submitted to a time- and concentration- dependent decrease. Our results show that the cytotoxicity of 3-ADON is lower than that of DON and 15-ADON at higher doses, but more than that of DON at 0.1 µM. Both 3-ADON (trichothec-9-en-8-one, 3-(acetyloxy)-12,13-epoxy-7,15-dihydroxy-, (3α,7α)) and 15-ADON (trichothec-9-en-8-one, 15-(acetyloxy)-12,13-epoxy-3,7-dihydroxy-, (3α,7α)) have the molecular formula C_17_H_22_O_7_, and a molecular weight of 338.35 g/mol [33]. The only distinction between them, is the position of the acetyl. Desjardins et al. found that C-15 esterification can increase toxicity (from DON to 15-ADON) [34]. Addition of an acetyl group at the R1 position (C3) of the molecule (3-ADON) resulted in a five- to more than 100-fold loss in potency when compared to the parent compound [35]. Since the relationship between cytotoxicity and chemical structure is complicated by the fact that toxicity in the cell line can be affected by complex interactions, the assay of cytotoxicity may have diverse results in different cells.

Based on the results of RNA-seq, we obtained a global view of the gene expression of Caco-2 cells under DON, 3-ADON and 15-ADON conditions. Substantial changes were noted between the toxin treated groups and control group. The number of upregulated genes and downregulated genes is almost the same. As shown in the Venn diagrams, more than half of the DEGs in each group were commonly orchestrated by all three toxins, which indicates that the toxic effects of the three toxins are analogous in Caco-2 cells.

All the DEGs were subjected to KEGG and GO analyses with different filter conditions. The enrichment analysis of the RNA-seq data showed that DON, 3-ADON and 15-ADON commonly affect a wide range of pathways, such as DNA replication, cell cycle, base excision repair, nucleotide excision repair, mismatch repair, drug metabolism via other enzymes and the p53 signaling pathways.

In the cell cycle pathway, the upregulation of ataxia-telangiectasia mutated kinase (ATM) and WEE1 homolog 2 (WEE2) were observed in all three groups. ATM exist as an inactive homodimer in undamaged cells until double-strand breaks (DSBs) or DNA replication blockage occurs. In such case, ATM will act as the initial and primary transducer of signaling [36,37]. Thus, the upregulation of ATM indicated that DNA damage in Caco-2 cells was induced by DON, 3-ADON and 15-ADON. In order to maintain genome stability, cells adopt multiple repair mechanisms and checkpoint responses that can delay cell cycle progression or modulate DNA replication. DNA damage can induce cell cycle delays at the G1/S and G2/M transitions (the G1 and G2 checkpoints) and also at the intro-S checkpoint [37]. In response to DSBs, ATM firstly activates checkpoint kinase 2 (CHK2), which in turn triggers the G1/S checkpoint by catalyzing the activating phosphorylation of the p53 [38]. Subsequently, p53 inhibits the CDK2-cyclin E complex by transactivating the cyclin dependent kinase (CDK) inhibitor p21 [39]. Apart from p53, CDK2 is also regulated by the cell division cycle 25A (CDC25A), the ATM-CHK2-CDC25A-CDK2 pathway is believed to control the S-phase checkpoint [40]. The low activity of CDK allows for an assembly of the pre-replication complex (pre-RC), a state competent for replication [41]. Apart from regulating the DNA replication directly, CDK-cyclin E can also collaborate with E2F for induction of the S-phase [42]. Thus, the downregulation of CDK and E2F indicate that the S-phase in the cell cycle was suppressed by DON, 3-ADON and 15-ADON. CDK1, governed by CDC25A and WEE1, is a pivotal regulator of the cell cycle throughout the S and G2-phase. In response to DNA damage, the upregulation of WEE1 was elicited to suppress the activity of CDK1/cyclin and induced cell cycle checkpoints during the S and G2-phases [43]. Many findings are consistent with our study on the toxic effects of DON and its acetylated derivatives which have been widely reported to cause DNA damage and inhibit the cell cycle [44,45,46,47].

The p53-regulated network can be divided into several different parts, each part regulates different functions in cells [48]. Apart from the correlation between p53 and the cell cycle, it is rather remarkable that sestrin 3 (SESN3) and SENE2, which are the target genes of DNA repair and damage prevention, were upregulated by DON, 3-ADON, 15-ADON and DON, 3-ADON in the present study. SESN2 is induced via genotoxic or oxidative stress mechanisms and SESN3 is activated in response to cell stimulation by serum or growth factors. SESNs exhibit antioxidant responses due to their oxido-reductase activity, thus, protecting cells from oxidative stress [48,49]. Based on the antioxidant characteristic of SESNs, a hypothesis can be confirmed that the DNA damage in Caco-2 cells was elicited by oxidative stress.

Upon both KEGG and GO enrichment, the most regulated pathway in three groups was found to be the DNA replication signaling pathway, which has an affinity with the G1 and S-phase of the cell cycle. In the KEGG analysis, all the genes enriched in DNA replication were downregulated, encompassing proliferating cell nuclear antigen (PCNA), origin recognition complex (ORC) and minichromosome maintenance (MCMs). The presence of conserved processes of eukaryotic DNA replication is coupled with the formation of pre-replication complexes (pre-RCs) either at, or very near to, the origin. This process involves the binding of the origin recognition complex (ORC) to DNA and the recruitment of cell division cycle 6 (Cdc6) and Cdt1-MCM (Mcm2–7). The formation of pre-RCs happens in the G1-phase of the cell cycle, when cells inter the S-phase, Dbf4-dependent kinase (DDK) will activate the formation of Cdc45/Mcm2–7/GINS (CMG) complex, to unwind the DNA. This allows the synthesis of the primer by the DNA polymerase α complex. PCNA is loaded at the primer template junction by the multi-subunit replication factor C (RFC) complex, which can open the PCNA ring and clamp it on the DNA. [50,51,52,53]. PCNA also works as an auxiliary protein of mammalian DNA polymerase δ (POLD) to stimulate the activity of the DNA polymerase δ core enzyme [54]. Additionally, PCNA may even dissociate and remain behind the fork to mark the leading strand for post-replicative events, such as mismatch repair (MMR), base excision repair (BER) and nucleotide excision repair (NER), which play key roles in maintain genome stability and the response to DNA damage [53]. In the present study, as a consequence of DNA damage, the three pathways have been affected to varying degrees. Parts of genes which participated in DNA synthesis, including PCNA, RFC, POLD, LIG1, were commonly downregulated in these pathways. Comparing this to the other two pathways, the toxic effects on MMR is weaker. MMR suppresses DNA recombination and plays a role in DNA damage signaling in mammalian cells. In the present study, ATPases MutSα and MutSβ, which play a documented role in recognition of mispairing DNA, were not significantly affected by DON, 3-ADON and 15-ADON. However, PCNA, RFC and EXO1, which regulate the 3′ nick-directed MMR, were observed to be downregulated [55]. NER is a highly conserved mechanism that can recognize and remove helical distortions throughout the genome [56]. RBX1, as a component of the recognition factors Cult4-DDB and Cul4-CSA, was downregulated by 3-ADON. TFIIH, which the NER machinery is built around, was downregulated by 3-ADON and 15-ADON. BER repairs damaged bases and small lesions from deamination, oxidation, or alkylation events [53,57]. In this pathway, PARP4 and NEIL1 were upregulated by DON, and 3-ADON, NEIL1 and OGG1 were upregulated by 15-ADON. NEIL1 and OGG1 are bifunctional DNA glycosylases that are involved in the first step of the Base excision repair (BER) pathway to remove modified DNA bases from damaged DNA [58]. The upregulation of NEIL1 and OGG1 may have a close relationship with oxidative damage to the bases [59]. It also means that the DSBs might be ascribed to oxidative stress, which is consistent with the conclusion mentioned above. However, further analysis is needed to verify this hypothesis. The suppression of DNA replication and the changes of DNA repair pathways certificate that DNA damage occurred in Caco-2 cells. This result is in agreement with that by Tiemann et al. [60].

In conclusion, the results mentioned above show that DON, 3-ADON and 15-ADON have analogous effects on Caco-2 cells. Although differences exist, the main toxic effects shown in our study are similar. Suppression of the cell cycle, DNA replication and DNA repair pathways occurred in all three groups. DON, 3-ADON and 15-ADON induce DNA damage and cause DSBs so that cell cycle arrest and DNA synthesis restraint can occur. The mechanisms that cause the DNA damage remain unclear, but according to the evidence we found, it can be assumed that oxidative stress is the culprit of the damage. This evidence shows that, even at low concentrations, the three toxins can exert severe influences, especially in the form of DNA damage, on Caco-2 cells. The results can provide a guideline for further experiments. Considering the noxious effects 3-ADON and 15-ADON can exert on animals and humans, more attention should dedicated to these compounds.

## 4. Materials and Methods

### 4.1. Reagents

Fatal bovine serum and Roswell Park Memorial Institute (RPMI) 1640 medium were purchased from Gibco by Thermo Fisher Scientific Inc. (Shanghai, China). Penicillin–streptomycin solution and 0.25% pancreatin were obtained from Genom (Hangzhou, China). DMSO was obtained from Sigma-Aldrich Inc. (Shanghai, China). Purified DON, 3-ADON, 15-ADON were purchased from Romer (Tulln, Austria).

### 4.2. Cell Culture and Treatment

The human intestinal Caco-2 cell line was obtained from China Center for Type Culture Collection. Caco-2 cells were cultured in RPMI1640 medium with 10% fetal bovine serum (FBS), 1% penicillin-streptomycin solution at 37 °C in a humidified atmosphere (95%) containing 5% CO_2_ in a CO_2_ incubator (Memmert, Germany).

### 4.3. Cell Viability Assay

Cell viability was determined by using a cell-counting kit-8 according to the manufacturer’s protocol (Shanghai, China). Cells (1 × 10^6^/mL) were seeded into a 96-well plate and cultured overnight, treated with different concentrations (0.1, 0.2, 0.5, 1, 2, 5, 10 µM) of DON, 3-ADON and 15-ADON for 24, 48, 72 h, respectively. Toxins were diluted with serum-free cell culture medium before use. The absorbance of the assay solution was measured at 450 nm by using a microplate reader (Perkin Elmer, Waltham, MA, USA). Results are expressed as a percentage relative to those for the controls. The viability was calculated using Equation (1):(1)$$\mathrm{Cell}\;\mathrm{viability}\;(\%)\;=\;\frac{As_{\left(treatment\right)}\;-\;Ab_{\left(blank\right)}}{Ac_{\left(control\right)}\;-\;Ab_{\left(blank\right)}}\times 100\%$$

### 4.4. RNA Extraction

Total RNA was extracted from Caco-2 cells (Appendix A) using Trizol reagent (Vazyme, Nanjing, China). The concentration and purity of the RNA samples were examined at 260/280 nm ratio by Nanodrop (Thermo Scientific, Waltham, MA, USA). RNA degradation and contamination were monitored on 1% agarose gel electrophoresis (AGE) by Mini-Sub Cell GT System (Bio-Rad, Minneapolis, MN, USA).

### 4.5. RNA-seq Analysis

Caco-2 cells under DON, 3-ADON and 15-ADON treated were placed in the frozen storage tubes and putted into liquid nitrogen container quickly. The samples were examined by CapitalBio Technology (Beijing, China). Differentially expressed genes (DEGs) were selected utilizing a cut-off threshold of |fold change| ≥ 2 and adjusted *p*-value ≤ 0.01. DEGs were analyzed using R language (R v 4.0.2, The R Foundation, Auckland, New Zealand, 2020). Pheatmap package in R was used to draw the heatmap of DEGs. Venn diagram was depicted by VennDiagram package and R. ClusterProfiler package in R was used for Kyoto Encyclopedia of Genes and Genomes (KEGG) and Gene Ontology (GO) enrichment analysis. The pathways considered to be significantly were choose with *p*-value ≤ 0.05, *q*-value ≤ 0.2 and *p*-adjust ≤ 0.01 and *q*-value ≤ 0.05, respectively. The details of the experiments can be found in GEO entry GSE164334, which also contains all the details of the experiments.

### 4.6. Real-Time PCR (RT-PCR) Assay

The expression of MCM3, MCM4, PCNA, POLE2, POLD2, RPA3, RAD51, BID, ATM, CDKN2C, E2F2, CCND2, SERPINE1 was measured by RT-PCR. RNA was synthesized cDNA using PrimeScriptTMRT Master Mix (TakaRa, Japan). The RT-PCR was conducted with CFX96 Real-Time PCR Detection System (Bio-rad, Minneapolis, MN, America). The thermal conditions were as follows: 95 °C for 30 s, 40 cycles of 95 °C for 5 s, 60 °C for 30 s. Primers were designed using the software primer 6 (Table 3). β-actin was used as an internal reference gene. The relative changes of mRNA expression were quantified by using the 2^−∆∆^CT method.

### 4.7. Statistical Analysis

All data are expressed as means ± standard error and were analyzed using IBM^®^ SPSS Statistics (Version 21.0, IBM corp., New York, NY, USA, 2012). Least significant difference (LSD) and Duncan were used for multiple comparisons. Significance was set at *p* < 0.05.

## Figures and Tables

**Figure 1 toxins-13-00167-f001:**
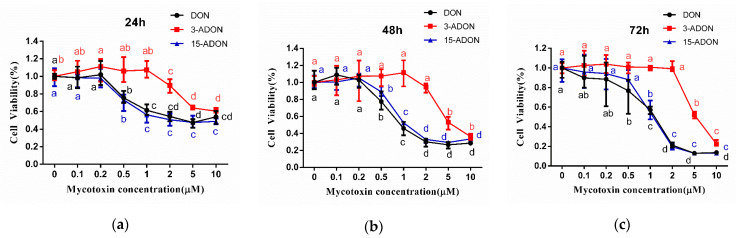
Cell viability in Caco-2 cells under deoxynivalenol (DON), 3-acetyldeoxynivalenol (3-ADON) and 15-acetyldeoxynivalenol (15-ADON) treatment. Results are expressed as mean ± SD (*n* = 5). Different letters (a, b, c) of same toxins are significantly different (*p* < 0.05). (**a**–**c**) represent 24 h, 48 h and 72 h exposed under deoxynivalenol (DON), 3-acetyldeoxynivalenol (3-ADON) and 15-acetyldeoxynivalenol (15-ADON) treatment respectively.

**Figure 2 toxins-13-00167-f002:**
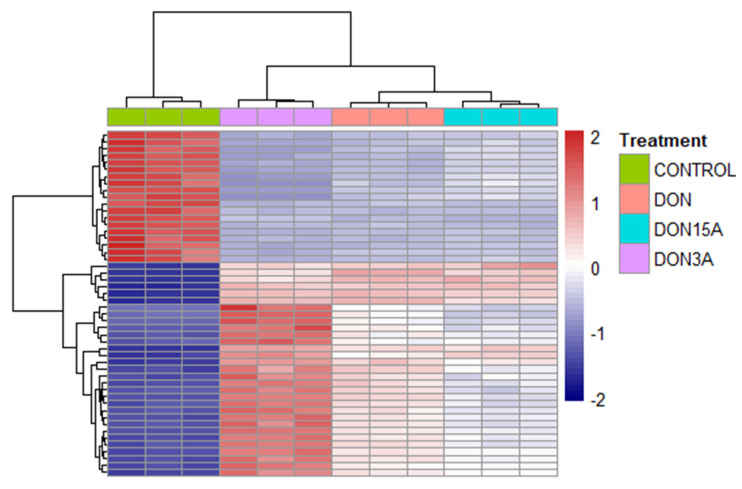
Transcriptomic changes of Caco-2 cells exposed to DON, 3-ADON or 15-ADON. Cluster heatmap representing differentially expressed genes between the control, DON, 3-ADON and 15-ADON conditions. Caco-2 were exposed to 0.5 µM DON, 3-ADON and 15-ADON for 24 h. The colors purple, green, orange and pink represent the control group, DON group, 3-ADON group and 15-ADON group, respectively.

**Figure 3 toxins-13-00167-f003:**
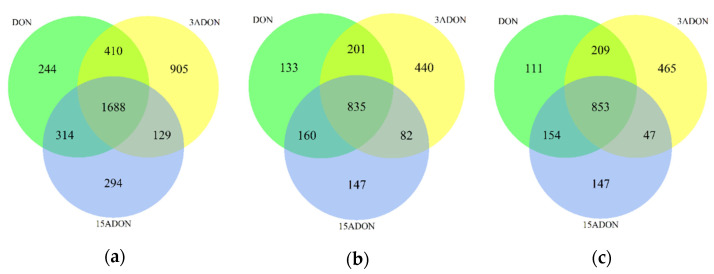
Venn diagram of differentially expressed genes (DEGs) in Caco-2 cells. (**a**) revealed that 1688 DEGs regulated by DON, 3-ADON and 15-ADON were all the same. (**b**) represents the relationship of up-regulated DEGs between the three groups and (**c**) represents the down-regulated DEGs.

**Figure 4 toxins-13-00167-f004:**
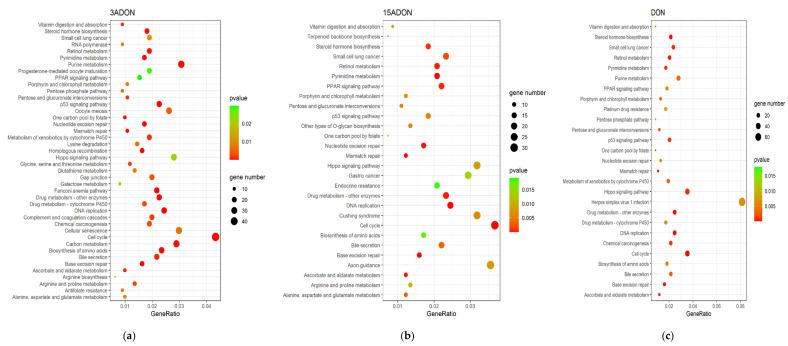
Bubbles of Kyoto Encyclopedia of Gene and Genome (KEGG) pathways of the DEGs. The size of bubbles represents the number of DEGs enriched in the pathway, the colors of bubbles indicate higher enrichment (lower *p*-value) in red. (**a**) 3-ADON treated group. Among the 42 pathways, cell cycle pathway was significantly influenced. (**b**) 15-ADON treated group. Among 28 pathways, cell cycle and DNA replication pathways were significantly affected. (**c**) DON treated group. Among 27 pathways, cell cycle and hippo signaling pathways were significantly affected.

**Figure 5 toxins-13-00167-f005:**
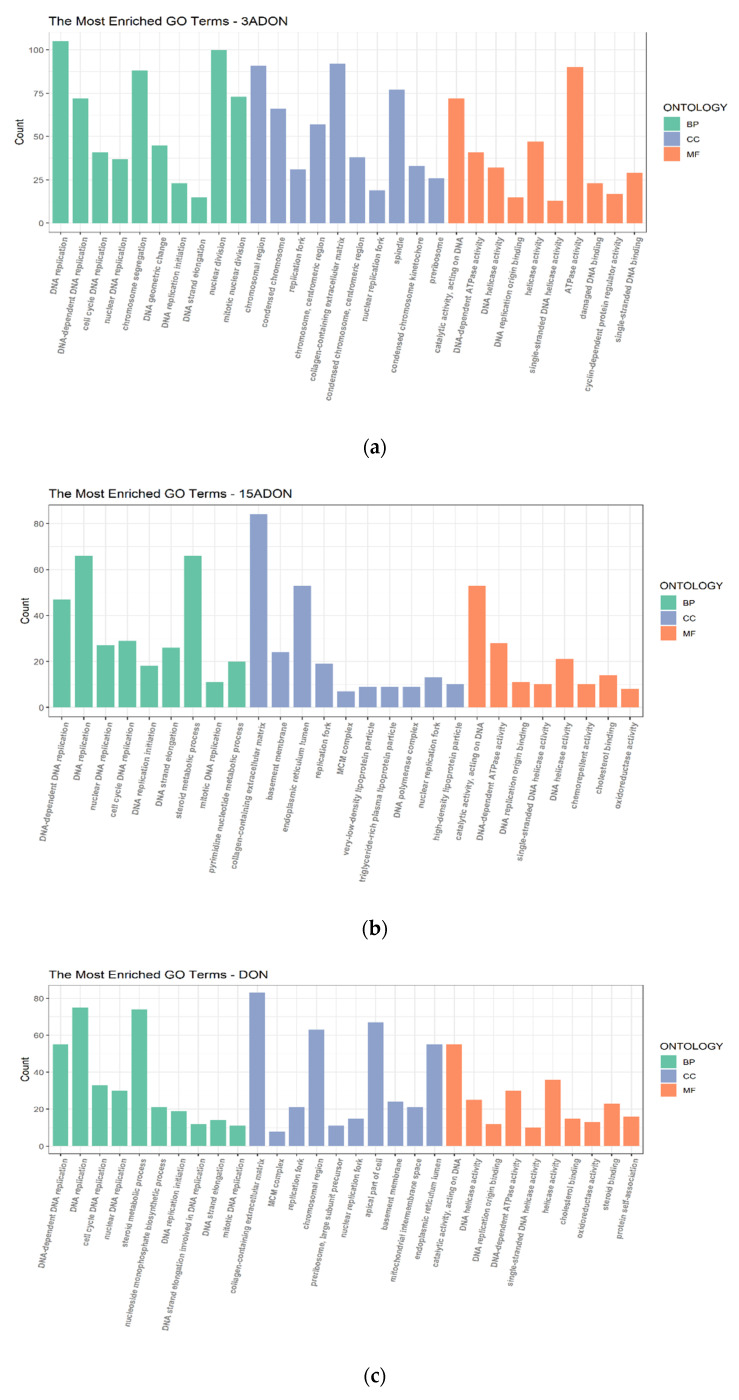
Gene Ontology (GO) enrichment analysis. Different colors represent different gene ontology: biological processes (BP), cellular components (CC) and molecular functions (MF). Top 10 pathways of each gene ontology were selected. (**a**) 3-ADON treated group. (**b**) 15-ADON treated group. (**c**) DON treated group.

**Figure 6 toxins-13-00167-f006:**
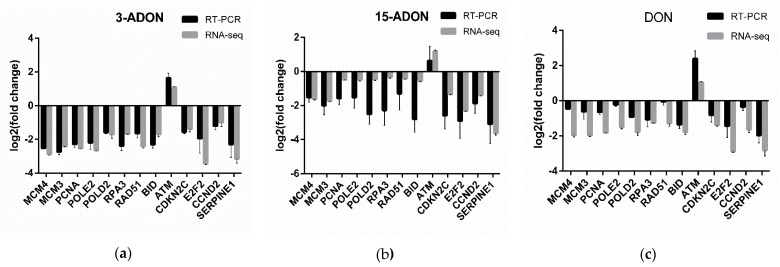
The expression changes of thirteen genes determined by Real-time PCR (RT-PCR) and RNA-seq. The x-axis represents the mRNAs and the y-axis is the log2 (fold change). Data represent mean ± SD, *n* = 3. (**a**) under 3-ADON treatment. (**b**) under 15-ADON treatment. (**c**) under DON treatment.

**Table 1 toxins-13-00167-t001:** IC_50_ of DON, 3-ADON and 15-ADON on Caco-2 cells after 24, 48 and 72 h of exposure.

Exposure Time (h)	IC_50_ (µM) ± SD		
	DON	3-ADON	15-ADON
24	6.17 ± 0.93	13.19 ± 0.71	3.86 ± 0.31
48	3.86 ± 2.21	10.81 ± 3.84	2.33 ± 0.36
72	1.46 ± 0.42	7.84 ± 2.01	1.44 ± 0.67

**Table 2 toxins-13-00167-t002:** The top 5 pathways of KEGG enrichment analysis results.

Toxin	Top 5 Pathways	Downregulated Genes	Upregulated Genes	*p*-Value	Count
3-ADON	DNA replication	PCNA, MCM4, MCM3, MCM5, MCM6, MCM7, MCM2, POLD1, RFC3, FEN1, POLE2, RFC4, RPA3, RNASEH2A, POLE3, LIG1, POLD3, PRIM1, POLA2, POLD2, RPA1, RFC5, POLE, RNASEH1, RFC2, POLA1, RNASEH2C	−	1.06 × 10^−16^	27
	Cell cycle	PCNA, MCM4, E2F1, MCM3, MCM5, CDC6, MCM6, MAD2L1, MCM7, MCM2, CDK1, CCNB1, PTTG1, E2F2, ORC6, CHEK2, CCNA2, CCND1, YWHAB, CDC45, PKMYT1, E2F3, ORC1, CDC25A, CCNE1, CDK2, CDKN2C, ESPL1, TFDP1, CHEK1, BUB1, DBF4, PLK1, RBX1, BUB1B, CDKN2A, CDC25C, MAD2L2, CDC7, CCNE2, SMC1B, CCND3, CDKN2D, CCND2	WEE2, STAG1, STAG2, ATM	2.74 × 10^−12^	48
	Base excision repair	PCNA, PARP1, POLD1, FEN1, POLE2, POLE3, UNG, LIG1, POLD3, NEIL3, NTHL1, POLE, PARP2, NEIL2, HMGB1, POLD2	NEIL1, PARP4	3.53 × 10^−8^	18
	Fanconi anemia pathway	RMI2, UBE2T, RAD51, CENPX, RPA3, EME1, RPA1, TELO2, FANCI, FANCA, FANCG, FANCD2, BRCA1, FANCM, BLM, FANCB, FAAP24, RAD51C, CENPS, CENPS-CORT, ATRIP	EME2, POLK, POLI	3.58 × 10^−8^	24
	Homologous recombination	POLD1, RAD51, RPA3, RAD54L, EME1, POLD3, RPA1, POLD2, XRCC2, BRCA1, RAD54B, XRCC3, RBBP8, BLM, BARD1, RAD51C	ATM, ABRAXAS1	2.31 × 10^−6^	18
DON	DNA replication	PCNA, MCM4, MCM3, MCM5, MCM6, MCM2, MCM7, POLD1, POLD2, RFC3, POLE2, RFC4, FEN1, RPA3, RNASEH1, POLE3, LIG1, POLD3, RNASEH2A, PRIM1, POLA2, RFC2, RNASEH2C	−	1.17 × 10^−13^	23
	Base excision repair	PCNA, PARP1, POLD1, UNG, POLD2, POLE2, FEN1, POLE3, LIG1, POLD3, NEIL2, HMGB1, NTHL1	PARP4, NEIL1	1.10 × 10^−6^	15
	Cell cycle	PCNA, E2F1, MCM4, MCM3, MCM5, E2F2, MCM6, MCM2, MCM7, CCND1, CDK2, MAD2L1, CDC6, ORC6, CCNE1, CDKN2C, TFDP1, CHEK2, CDC45, E2F3, ESPL1, PKMYT1, CDC25A, ORC1, CCND3, CDKN2A, MAD2L2, CDKN2B, CCND2	WEE2, STAG1, STAG2, ATM	2.69 × 10^−6^	33
	Steroid hormone biosynthesis	CYP1A1, UGT1A6, HSD17B8, HSD11B2, UGT1A4, UGT1A1	UGT2A3, UGT2B15, UGT2B17, AKR1C2, UGT2B7, UGT2B11, STS, AKR1C1, AKR1C3, CYP17A1, CYP3A5, SULT1E1, UGT2B10, AKR1D1	9.00 × 10^−6^	20
	Drug metabolism—other enzymes	TK1, NME1, NME4, DUT, NME2, HPRT1, IMPDH1, UCK2, UGT1A6, UPP1, TK2, MGST1, UGT1A4, CDA, UGT1A1	UGT2A3, GSTA2, UGT2B15, UGT2B17, UGT2B7, UGT2B11, GSTA1, UGT2B10	1.81 × 10^−5^	23
15-ADON	DNA replication	PCNA, MCM3, MCM4, MCM5, POLD1, MCM6, MCM2, MCM7, POLE2, POLD2, RFC4, RFC3, RPA3, LIG1, POLE3, RNASEH1, PRIM1, POLD3, POLA2, RFC2	−	1.54 × 10^−11^	20
	Cell cycle	PCNA, MCM3, E2F1, MCM4, MCM5, E2F2, CDK2, TFDP1, MCM6, CCND1, MAD2L1, MCM2, MCM7, ANAPC5, CCNE1, CDKN2C, CHEK2, E2F3, ORC6, CDC45, ORC1, PKMYT1, CDC25A, MAD2L2, CCND3, CCND2	WEE2, STAG1, ATM, CDKN2B	4.03 × 10^−6^	30
	Base excision repair	PCNA, PARP1, UNG, POLD1, POLE2, POLD2, LIG1, POLE3, POLD3, NTHL1, NEIL2	NEIL1, OGG1	8.87 × 10^−6^	13
	Pyrimidine metabolism	TYMS, NME1, TK1, NME4, DUT, DTYMK, UCK2, CAD, NME2, DCK, UPP1, TK2, DCTPP1, DHODH, CDA	ENTPD8, AK9	2.92 × 10^−5^	17
	Mismatch repair	PCNA, POLD1, POLD2, RFC4, RFC3, EXO1, RPA3, LIG1, POLD3, RFC2	−	3.57 × 10^−5^	10

“−” indicates no upregulated genes were found in corresponding pathways.

**Table 3 toxins-13-00167-t003:** Primer sequences of RT-PCR target genes.

Gene	Length (bp)	Primer Sequence
β-actin	19	F: CCTTCCTGGGCATGGAGTC
	21	R: TGATCTTCATTGTGCTGGGTG
MCM3	19	F: AAGCAGATGAGCAAGGATG
	19	R: CAAGAGCAAGCAGAGGATT
MCM4	17	F: CACCTGGTCGCACTGTA
	17	R: GGCTGGCTTCCTCACTT
PCNA	22	F: ACACTAAGGGCCGAAGATAACG
	22	R: ACAGCATCTCCAATATGGCTGA
POLE2	17	F: TGGTGGAAGCAGCAGTC
	21	R: GGTTGGTCATTAACAGAGGAA
POLD2	17	F: CTGGTGGATGTGGTGAC
	17	R: CTGTGGCTGAGGAGGTT
RPA3	21	F: AGCTCAATTCATCGACAAGCC
	22	R: TCTTCATCAAGGGGTTCCATCA
RAD51	22	F: CAACCCATTTCACGGTTAGAGC
	21	R: TTCTTTGGCGCATAGGCAACA
BID	17	F: CGTCCTTGCTCCGTGAT
	18	R: TGTCCGTTCAGTCCATCC
ATM	19	F: ACTACTGCTCCAGACCAAT
	20	R: TCACGACGATACAAAGAACA
CDKN2C	15	F: CTGAGCGGCATTAGC
	15	R: CGAACGGGAGTAGCA
E2F2	18	F: GCACTGGCATCATTCTCT
	18	R: AGTCACCTCTGTCCTTGG
CCND2	19	F: ACCTTCCGCAGTGCTCCTA
	19	R: CCCAGCCAAGAAACGGTCC
SERPINE1	17	F: GCTGGTGCTGGTGAATG
	17	R: AGTGCTGCCGTCTGATT

## Data Availability

Data are available in a publicly accessible repository. The data presented in this study are openly available in GEO database, reference number GSE164334.

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
