# Peer review of "Transcriptome Analysis of Caco-2 Cells upon the Exposure of Mycotoxin Deoxynivalenol and Its Acetylated Derivatives"

_toxins, 2021, doi:10.3390/toxins13020167_

Round 1

Reviewer 1 Report

General description

This study used RNA-seq as the main tool to investigate differentially expressed genes (DEGs) of a human colorectal tumor cell lines (caco-2) in response to the challenge of three mycotoxins, i.e., DON, 3-ADON and 15-ADON. Pathways associated with DNA damage, DNA replication and the regulations of oxidative stress were found induced by the challenge of these mycotoxins.

Major concerns:

This is an interesting study and, with more mycotoxins being identified, it is important to characterize their toxic effects on food and feeds. However, the present study is weak in its novelty and the objective is not clearly described.

Several concerns from this reviewer are described as follows.

  1. As mentioned by the authors, 3-ADON and 15-ADON are two less frequently found toxins and are account for just a small proportion of DON contaminations (~10%) in food and feeds. It appears that the authors had made it a big deal to study these relatively less health-threatening toxins, although it still has some value in a general sense of scientific research.
  2. As the main theme of this study, general symptoms or syndromes caused by these toxins in humans should be adequately described before looking into the gene expression profile in response to toxin challenges. Without this, the main objective of this study would become weak and blurred. After that, the DEG analysis would become much more meaningful.
  3. It would have been much powerful, instead, if at least two different cell types either human or animal cells were compared in this study. In addition, the use of Caco-2 cell line (the switch from animals to humans) needs to be justified, e.g., in the Introduction, because most, if not all, syndromes or disorders reported are on animals; Also, it would be better if a non-tumorigenic cell type, like IPEG-J2 cells for instance, were used as a comparison, which might help authors to generalize the phenomena learned from their DEG data in this study.
  4. Given that 3-ADON was found less toxic than DON and 15-ADON as shown in the cell viability assay, the author should have discussed more on comparing different cell types or cellular responses between various treatments based on the DEGs found in transcriptome analysis. Instead of validating the toxic effect on acetylated DONs, it would be more interesting to discuss the differential toxicity between 3-ADON versus DON and 15-ADON on their structural differences.
  5. This reviewer is not impressed that the DEG profiles (either up-regulation or down-regulation) are mostly associated with apoptosis, cell cycle arrest, induced DNA damage, or inhibition of DNA replication, and so forth, in Caco-2 cells which have been well-documented. In addition to those, it is critical for the authors to identify any new finding from their study.
  6. Concentrations and time of treatment durations selected for this study also have to be justified.

Specific/minor points:

The title description is unclear and has room to improve. A suggested title FYI:

“Transcriptome analysis of Caco-2 Cells upon the exposure of mycotoxin deoxynivalenol and its acetylated derivatives”.

Most results were not well-presented with clear explanation and/or footnotes. For example, abbreviations of gene ontology (BP, CC, and MF) should be annotated in figure legends (L156).

The statistical analysis of cell viability was also conducted within the same concentration in order to know the different effect between treatments.

L25 and L38: All fungal species should be italicized.

The first appeared abbreviation should be spelled-out. e.g.,

L36: JECFA and PMTDI; L40: 3-acetyl-DON and 15-acetyl-DON; L57: ER stress.

L68: change “to analyzed” to “to analyze”.

What are the mean concentrations of 3-DON and 15-DON in the food and animal feeds?

Figure 1 shows that the cell viability of Caco-2 cells has a time- and dose-dependent decrease. Any statistical differences between different toxins at the same concentration and time?

L77: Change “IC50” to “IC50” and be consistent; also “24h, 48h, 72h” should be “24 h, 48 h, 72 h” and be consistent. Please check through the text carefully.

Figures 3-6: Please briefly explain the meaning of each plot, i.e., (a), (b), and (c).

Table 2: Please check again the expression of P value.

The company information of chemicals should be correct enough and complied with the guides for the authors of this journal; For examples,

L270: Is the location of Thermo Fisher Scientific in the USA or in China?

L273: Should the correct name of company be “Romer Lab”?

L281: Is the location of Dojindo in Japan or in China?

L290, 292, and 309: Please change “America” to “USA” for 4.1 to 4.3 being “USA”. Be consistent throughout; also Bio-rad or Bio-Rad ?

The descriptions in the materials and methods for cell culture are too simplified, if I didn’t miss any:

(1)    The compositions of the growth medium were not mentioned. What percentage of FBS was added to RPMI 1640 medium?

(2)    The procedure for diluting toxins into culture media requires more details

The subtitles used in the Results can be neutral or more informative. For example, I would not be so surprised when I read “DON, 3-ADON and 15-ADON inhibit the viability of Caco-2 cells” in section 2.1. Instead, it would be more informative if it were, for instance, “3-ADON exerts less harmful effect than DON and 15-ADON on cultured Caco-2 cells”. Otherwise, please use a neutral subtitle to describe your results, as in Sec 2.2. and 2.3. FYI.

Reviewer 2 Report

Very interesting and valuable paper comparing toxic mechanism of DON and their two derivatives.

It is important to know toxic effects of 3Ac-DON and 15AC-DON because they are produced by different F. graminearum chemotypes and frequency of these chemotypes in different countries/regions varies. It results in different amount of 3Ac-DON and 15-AcDON in cereal grain.

Interesting results is that 3Ac-DON is less toxic than 15-AcDON. However, authors did not try to explain it in the discussion.

This is the only thing I miss in this article. It can be published in present form but it would be more valuable with the above.

Reviewer 3 Report

Line 5: change „major“ to „one of major“

Line 24 – 25 please use italic when naming fungal species, and change the sentence from „Mycotoxins are toxic secondary metabolites produced by a few fungal species belonging mainly to the Aspergillus, Penicillium and Fusarium genera [1]“ to Mycotoxins are toxic secondary metabolites produced by a few fungal species belonging mainly to the Aspergillus, Penicillium, Fusarium and Alternaria genera [1, https://doi.org/10.2478/aiht-2018-69-3108]

Line 36-48: please mention also other DON metabolites that co-occur in food and feed, such as DON-3-Glc, DOM-1 [ref 5], DON sulfates [https://doi.org/10.1007/s00216-014-8340-4], DON sulfonates (https://doi.org/10.1007/s12550-019-00385-5); iso DON, norDON A, norDON B, norDON C [https://dx.doi.org/10.3390%2Ftoxins11060317] that have not been studied in details yet. And also other mammalian metabolites such as DON-3-GlcA, DON-15-GlcA [ http://dx.doi.org/10.1016/j.fct.2013.08.043]; iso-DON glucuronides and iso-deepoxy-DON/DOM glucuronides [https://doi.org/10.1007/s00204-017-2012-z]. Also please comment on the effect of co-contamination with other mycotoxins on toxicity/metabolism of DON (eg. Culmorin: https://doi.org/10.1007/s00204-019-02459-w), that often co-occur [http://dx.doi.org/10.1016/j.foodchem.2017.02.115].

Line 48-49: Please add some examples co-occurence of DON and its metabolites in real food & feed samples (eg. barley: http://dx.doi.org/10.3390/microorganisms7110532; wheat: http://dx.doi.org/10.3390/microorganisms8040578; https://doi.org/10.3390/toxins11040198; complementary foods/ infant food: https://doi.org/10.1016/j.foodcont.2018.11.049; https://doi.org/10.1016/j.fct.2018.08.025; beer: https://doi.org/10.1016/j.foodchem.2018.02.005; https://doi.org/10.1080/19440049.2012.726745)

Line 49 Change to The toxicology of DON is well documented.

Figure 3 – please explain what do numbers represent in the figure description

Line 270 Please add the city of Gibco, and Sigma-Aldrich (line 272)

Line 273 Romer headquarters are in Tulln, Austria

Line 276 please correct °C, (also in rest of the text (eg. Line310) add humidity in the incubator

Line 292 can you add RNA degradation gels images in supplementary files

Round 2

Reviewer 3 Report

The authors have corrected the manuscript appropriately. There are still some minor issues to be corrected prior the publishing.

line 43. please add DON sulfates as one of the plant metabolites of DON (reference: https://doi.org/10.1007/s00216-014-8340-4)

line 57. please add infant food (https://doi.org/10.1016/j.foodcont.2018.11.049) and beer (https://doi.org/10.1016/j.foodchem.2018.02.005; https://doi.org/10.1080/19440049.2012.726745) as relevant sources of DON and its metabolites to humans

Supplementary file. Please label the gel and add a brief explanation since it is hard to recognize what do we need to look in those bands. Is there a need for the red markings on the first two samples?)
